# Sporadic inclusion body myositis-derived myotube culture revealed muscle cell-autonomous expression profiles

**Naoki Suzuki**[1,2]*, **Makoto Kanzaki**[3], **Masashi Koide**[4], **Rumiko Izumi**[1], **Ryo Fujita**[4], **Tadahisa Takahashi**[4], **Kazumi Ogawa**[1,4], **Yutaka Yabe**[4], **Masahiro Tsuchiya**[5], **Masako Suzuki**[1], **Ryuhei Harada**[1], **Akiyuki Ohno**[1], **Hiroya Ono**[1,6], **Naoko Nakamura**[1], **Kensuke Ikeda**[1], **Hitoshi Warita**[1], **Shion Osana**[7], **Yoshitsugu Oikawa**[8,9], **Takafumi Toyohara**[9,10], **Takaaki Abe**[9,10,11], **Muliang Rui**[2], **Satoru Ebihara**[2], **Ryoichi Nagatomi**[7], **Yoshihiro Hagiwara**[4], **Masashi Aoki**[1]*

1 Department of Neurology, Tohoku University Graduate School of Medicine, Sendai, Japan, 2 Department of Rehabilitation Medicine, Tohoku University Graduate School of Medicine, Sendai, Japan, 3 Graduate School of Biomedical Engineering, Tohoku University, Sendai, Japan, 4 Department of Orthopedic Surgery, Graduate School of Medicine, Tohoku University, Sendai, Japan, 5 Department of Nursing, Tohoku Fukushi University, Sendai, Japan, 6 Department of Neurology, National Hospital Organization Iwate Hospital, Ichinoseki, Iwate, Japan, 7 Division of Biomedical Engineering for Health and Welfare, Graduate School of Biomedical Engineering, Tohoku University, Sendai, Japan, 8 Department of Pediatrics, Tohoku University Graduate School of Medicine, Sendai, Japan, 9 Division of Nephrology, Endocrinology and Vascular Medicine, Tohoku University Graduate School of Medicine, Sendai, Japan, 10 Department of Medical Science, Tohoku University Graduate School of Biomedical Engineering, Sendai, Japan, 11 Department of Clinical Biology and Hormonal Regulation, Tohoku University Graduate School of Medicine, Sendai, Japan

* aokim@med.tohoku.ac.jp (MA); naoki.suzuki.e3@tohoku.ac.jp (NS)

**Data Availability Statement:** The accession number for the RNA-seq data reported in this paper is DDBJ (DNA DataBank of Japan), https://

## Abstract

Sporadic inclusion body myositis (sIBM) is a muscle disease in older people and is characterized by inflammatory cell invasion into intact muscle fibers and rimmed vacuoles. The pathomechanism of sIBM is not fully elucidated yet, and controversy exists as to whether sIBM is a primary autoimmune disease or a degenerative muscle disease with secondary inflammation. Previously, we established a method of collecting CD56-positive myoblasts from human skeletal muscle biopsy samples. We hypothesized that the myoblasts derived from these patients are useful to see the cell-autonomous pathomechanism of sIBM. With these resources, myoblasts were differentiated into myotubes, and the expression profiles of cell-autonomous pathology of sIBM were analyzed. Myoblasts from three sIBM cases and six controls were differentiated into myotubes. In the RNA-sequencing analysis of these "myotube" samples, 104 differentially expressed genes (DEGs) were found to be significantly upregulated by more than twofold in sIBM, and 13 DEGs were downregulated by less than twofold. For muscle biopsy samples, a comparative analysis was conducted to determine the extent to which "biopsy" and "myotube" samples differed. Fifty-three DEGs were extracted of which 32 (60%) had opposite directions of expression change (e.g., increased in biopsy vs decreased in myotube). Apolipoprotein E (apoE) and transmembrane protein 8C (TMEM8C or MYMK) were commonly upregulated in muscle biopsies and myotubes from sIBM. ApoE and myogenin protein levels were upregulated in sIBM. Given that enrichment analysis also captured changes in muscle contraction and development, the triggering

ddbj.nig.ac.jp/resource/sra-submission/
DRA017123, DRA accession number: DRA017123.

**Funding:** This research was partially supported by Intramural Research Grants 29-4 and 2-5 for Neurological and Psychiatric Disorders provided to M.A. from the National Center of Neurology and Psychiatry of Japan; the Practical Research Project for Rare/Diseases (20dk0310086) provided to M.A. and Moonshot R&D Program (JPMJMS 23zf0127001h0003) to T.A. from the Japan Agency for Medical Research and Development (AMED); Grants-in-Aid for Research on Rare and Intractable Diseases (H29-nanchitou(nan)-ippan-030 and 20CF1036) provided to M.A. from the Ministry of Health, Labor and Welfare of Japan; a Grant-in-Aid for Challenging Exploratory Research (20K21563) provided to M.A. and N.S., Scientific Research C (18K07519) provided to N.S., from the Japanese Ministry of Education, Culture, Sports, Science and Technology. This research was also supported by the Cooperative Research Project Program of the Joint Usage/Research Center at the Institute of Development, Aging and Cancer, Tohoku University.

**Competing interests:** The authors have declared that no competing interests exist.

of muscle atrophy signaling and abnormal muscle differentiation via MYMK or myogenin may be involved in the pathogenesis of sIBM. The presence of DEGs in sIBM suggests that the myotubes formed from sIBM-derived myoblasts revealed the existence of muscle cell-autonomous degeneration in sIBM. The catalog of DEGs will be an important resource for future studies on the pathogenesis of sIBM focusing on primary muscle degeneration.

## Introduction

Sporadic inclusion body myositis (sIBM) is a defined muscle disease prevalent in older individuals [1–3], characterized by specific patterns of muscle involvement, including finger flexors, knee extensors, and dysphagia [4]. Pathologically, sIBM is identified by the presence of rimmed vacuoles and inflammatory cell invasion in non-necrotic muscle fibers [5]. Despite these distinctive features, the underlying pathomechanism of sIBM remains incompletely understood.

Controversy surrounds whether sIBM is a primary autoimmune disease or a degenerative muscle disorder with secondary inflammation. Reports suggest the involvement of the interferon-gamma mediated signaling pathway [6], myeloid dendritic and plasma cells [7], and highly differentiated cytotoxic T cells [8] in sIBM pathology. Additionally, anti-cytosolic 5'-nucleotidase 1A autoantibodies are found in the serum of patients with sIBM [9]. However, the disease often proves resistant to immunosuppressive therapy, emphasizing the importance of muscle degeneration in sIBM etiology [10].

Muscle biopsy is necessary for the diagnosis of sIBM [11, 12]. The rest of the diagnostic samples can be used for research, such as in gene expression profiling; for example, gene sets to differentiate sIBM biopsy samples from other myositis using support vector machine-learning system are revealed [13]. Other groups analyzed two microarray datasets [8, 14] derived from the Gene Expression Omnibus database and picked-up genes as a hub of a pathological network [15]. As the pathology of sIBM shows the involvement of cells such as inflammatory cell infiltration, fibrosis, and adipose tissues [16, 17], challenges arise due to the potential confounding effects of non-muscle cells in expression data analysis.

A significant development in understanding sIBM occurred in 1994 when ApoE immunoreactive deposits were first identified in rimmed vacuoles of sIBM muscle biopsy samples [18]. Subsequent findings implicated aberrant cholesterol metabolism in sIBM muscles [19, 20]. An sIBM mouse model overexpressing amyloid beta was utilized [21], yet the mechanisms of abnormal protein accumulation in muscle degeneration remain unclear.

Previously, we established a method of collecting CD56-positive myoblasts from human skeletal muscles derived from muscle biopsy samples [22, 23]. We hypothesized that the myoblasts could serve as a valuable resource to investigate the cell-autonomous pathomechanism of sIBM. The myoblasts were differentiated into myotubes, and the expression profiles were analyzed to gain insights into the cell-autonomous pathology of sIBM.

## Materials and methods

### Study population

The study protocol was approved by the Tohoku University Hospital's Institutional Review Board (Approval nos. 2014-1-703, 2016-1-884, 2019-1-493, 2022-1-848), and written informed consent was obtained from all participants. Data was collected on 12/10/2020.

**Table 1. Characteristics of patients with sporadic inclusion body myositis.**

| Case No. | Age at biopsy | Sex | Diagnosis | muscle ID | Site of biopsy | Myoblast ID | Biopsy RNA-seq |
|---|---|---|---|---|---|---|---|
| 1 | 86 | F | sIBM | C1017 | BB | sIBM Myoblast 2 | |
| 2 | 69 | M | sIBM | C1047 | RF | sIBM Myoblast 6 | sIBM Biopsy 1 |
| 3 | 69 | M | sIBM | C1054 | BB | sIBM Myoblast 7 | sIBM Biopsy 2 |
| 4 | 73 | M | sIBM | C1092 | VL | | sIBM Biopsy 3 |
| 5 | 62 | M | sIBM | C1220 | BB | sIBM Myoblast 8 | |
| 6 | 77 | M | neurogenic atrophy | C1053 | BB | | Disease control Biopsy 1 |
| 7 | 65 | F | neurogenic atrophy | C922 | Gc | | Disease control Biopsy 2 |
| 8 | 71 | F | DMD carrier | C1032 | BB | | Disease control Biopsy 3 |
| 9 | 76 | M | amyloidosis | C1049 | RF | | Disease control Biopsy 4 |
| 10 | 56 | F | Rotator cuff tears | | subscapularis | SSC13 | |
| 11 | 60 | F | Rotator cuff tears | | subscapularis | SSC14 | |
| 12 | 69 | F | Rotator cuff tears | | subscapularis | SSC16 | |
| 13 | 60 | M | Rotator cuff tears | | subscapularis | SSC18 | |
| 14 | 69 | M | Rotator cuff tears | | subscapularis | SSC4 | |
| 15 | 69 | M | Rotator cuff tears | | subscapularis | SSC8 | |

BB; biceps brachii, DMD; Duchenne muscular dystrophy, Gc; Gastrocnemius, RF; rectus femoris, VL; vastus lateralis

## Sample collection from patients with sIBM

From 26/01/2015 to 31/05/2018, we performed muscle biopsies on 13 patients who were presumptively diagnosed with sIBM. Muscle samples were collected from nine patients who agreed to undergo a biopsy. The number of patients evaluated was restricted because of the limited access to the fluorescence-activated cell sorting (FACS) system at our institute and the requirement for performing the procedure on the same day as the muscle biopsy.

The cells prepared from each muscle by FACS were separated into three replicates for characterization and culture. In each patient, muscle biopsy specimens (approximately 300 mg) were obtained from the muscle belly of the biceps brachii or lower proximal muscles (Table 1). Three lines of actively proliferating myoblasts were used for the analysis.

Sample collection from the control participants (patients with rotator cuff tears [RCT]).

From 26/01/2015 to 28/12/2015, arthroscopic RCT repairs were performed in 42 patients who were unresponsive to conservative treatments [23]. Muscle samples were collected from 19 patients who agreed to undergo a biopsy. None of the patients had tears in the intact subscapularis (SSC) tendon based on both magnetic resonance imaging and arthroscopic findings. No patients had muscular disease, neurovascular disorders, paralysis, or trauma. Muscle biopsy specimens (approximately 300 mg) were obtained from the same portion of the muscle belly of the SSC in each patient who underwent arthroscopic surgery (Table 1) [23].

## Primary myoblast isolation

Human satellite cells were isolated from the muscles of patients who agreed to undergo biopsy. All experiments were performed in accordance with relevant guidelines and regulations as described elsewhere [23, 24]. Briefly, the tissue was minced and digested with 0.2% collagenase (Wako Pure Chemicals Industries) and 0.1% DNase I (Sigma-Aldrich, St. Louis, MO, USA), filtered through a 70-μm cell strainer (BD Biosciences, Franklin Lakes, NJ, USA), and centrifuged at $700 \times g$ for 20 min. Pellets were resuspended in phosphate-buffered saline (PBS) containing 1% bovine serum albumin (BSA; Sigma-Aldrich) and then incubated with an Fc

receptor blocking solution (Human TruStain FcX, 1:20 in the staining buffer; BioLegend, San Diego, CA, USA). Then, the samples were labeled with the following monoclonal antibodies (all from BioLegend and all at 1:20 dilution): fluorescein isothiocyanate (FITC)-conjugated anti-CD45 (clone HI30), FITC-conjugated anti-CD11b (clone ICRF444), FITC-conjugated anti-CD31 (clone WM59), phycoerythrin (PE)/Cy7-conjugated anti-CD34 (clone 581), allophycocyanin (APC)-conjugated anti-CD56 (clone MEM-188), and PE-conjugated anti-PDGFRα (clone 16A1). The negative set included blood markers CD11b and CD45 and endothelial markers CD31 and CD34. Although CD34 is expressed by the majority of mouse satellite cells [25], human muscle-derived CD34$^+$ cells are myogenic and adipogenic, whereas CD34$^-$ cells are myogenic but not adipogenic [26]. Therefore, CD34 was used as a negative selection marker. Human satellite cells were defined as single live mononuclear CD11b$^-$CD31$^-$CD34$^-$CD45$^-$CD56$^+$ cells. FACS was performed on a FACS ARIA II flow cytometer (BD Biosciences). Cells were seeded onto 24-well chamber slides coated with Matrigel (Dow Corning, Corning, NY, USA) in a growth medium containing DMEM/Ham's F10 mixture supplemented with 20% fetal bovine serum, 1% penicillin—streptomycin, 1% chicken embryonic extract (United States Biological, Salem, MA, USA), and 2.5 ng/mL basic fibroblast growth factor (Thermo Fisher Scientific, Waltham, MA, USA) and cultured at 37°C in a 5% CO$_2$ atmosphere. When cells reached 60%–80% confluence, adherent cells were dissociated and split onto a new Matrigel-coated 15-cm dish to expand the activated satellite cells. Activated satellite cells (myoblasts) were suspended in Cell Banker (Takara, CB011, Japan) and stored in liquid nitrogen.

## Cell culture

Human myoblasts were cultured using media purchased from Lonza (Walkersville, MD, USA). Three days after plating, the cells reached 80%–90% confluence (day 0). Then, differentiation was induced by switching the growth medium to DMEM supplemented with 2% horse serum, 30 μg/mL penicillin, and 100 μg/mL streptomycin (differentiation medium). The differentiation medium was changed every 24 h during the 7–8 days of differentiation.

## RNA extraction and sequencing analysis

Cultured myotubes were collected to extract the total RNA, which was performed using an RNeasy micro kit (Qiagen, Germany) according to the manufacturer's protocol. RNA-sequencing (RNA-seq) libraries were prepared using a TruSeq RNA-stranded mRNA Sample Prep Kit (Illumina, CA, USA). These libraries were clonally amplified on a flow cell and sequenced on a HiSeq2500 (HiSeq Control Software v2.2.58, Illumina) with a 51-mer single-end sequence. Image analysis and base calling were performed using Real-Time Analysis Software (v1.18.64, Illumina). For data analysis, UCSC hg19 and RefSeq were used as the reference human genome and gene model, respectively. For gene expression analysis, single-end reads were mapped to the human genome using TopHat (ver. 2.1.0) [27]. Cufflinks (ver. 2.2.1) was used to estimate the gene expression levels based on fragments per kilobase of the exon model per million mapped fragments [28]. Gene expression levels were compared between control cells and sIBM cells using Cuffdiff (ver. 2.2.1). The False Discovery Rate (FDR) cutoff value was set at 0.05. Unmapped reads are 4–7% % (S1 Table).

## Immunohistochemistry and immunocytochemistry

The biopsy sections and cultured myotubes were washed with PBS and fixed for 20 min with 2% paraformaldehyde in PBS containing 0.1% Triton X-100. Samples were washed and blocked in PBS containing 5% CS and 1% BSA at room temperature. For immunofluorescence

analysis, anti-desmin antibody (MAB-606102, Diagnostic BioSystems: DBS), ApoE (16H22L18, Invitrogen), dystrophin (NCL-Dys1, Leica), caveolin-3 (sc-55518, Santa Cruz), MYMK (PA5-63180, Invitrogen), amyloid oligomers (A11; SPC-506D, StressMarQ), were used as the first antibody, and Alexa Fluor 488 or 568-conjugated anti-IgG was used as the secondary antibody, in a solution of 1% BSA in PBS. The samples were mounted on glass slides with Vectashield (Vector Laboratories, Burlingame, CA, USA) and observed with a confocal fluorescence microscope (Fluoview FV-1000; Olympus, Tokyo, Japan) or fluorescence microscope (BZ-X700; Keyence).

## Immunoblotting

For immunoblot analysis, skeletal muscle protein was extracted from the human myotubes, as described previously [29]. Total cell proteins were extracted from cells with radio-immunoprecipitation (RIPA) buffer (ATTO) and measured using a Bicinchoninic Acid Kit (Thermo Fisher Scientific). After the adjustment of the protein concentration, protein samples were reacted with Laemmli buffer at 95˚C for 5 min and separated by sodium dodecyl sulfate-polyacrylamide gel electrophoresis using 10%–20% polyacrylamide gels (ATTO) under constant voltage of 200 V for 60 min. Then, the separated proteins were transferred to polyvinylidene difluoride membranes (Millipore, MA, USA). After blocking with 2% BSA (Pierce), membranes were probed with the indicated primary antibodies including ApoE (16H22L18, Invitrogen), myogenin (sc-12732, Santa Cruz), and GAPDH (2118L, CST) overnight at 4˚C. Immunolabeling was visualized by incubation with horseradish peroxidase-linked secondary antibodies (5,000-fold dilution) at room temperature for 1 h, followed by treatment with an enhanced chemiluminescence prime reagent (GE Healthcare, IL, USA); images were captured using an LAS-3000 Image Reader (Fujifilm, Tokyo, Japan). Blots were labeled with anti-GAPDH antibodies as the gel loading control.

## Statistical analysis

Statistical analyses were performed using Student's t-test, analysis of variance with Tukey's multiple comparison test, or Wilcoxon signed-rank test, and p-values $< 0.05$ indicated a statistically significant difference unless otherwise specified. Data are expressed as means $\pm$ SE unless otherwise specified.

## Results

Myoblasts from three sIBM subjects and six controls were subjected to differentiation, resulting in the formation of myotubes (referred to as "myotube": Table 1 and Fig 1A).

Immunocytochemistry analysis demonstrated the presence of desmin-positive multinucleated myotubes following the differentiation process in both the sIBM and control groups (Fig 1B–1G). When quantified, it was determined that 95–98% of the nuclei exhibited colocalization with Desmin, as depicted in Fig 1H. Subsequently, RNA was extracted from these myotubes (Fig 1A), and RNA-seq was conducted (Fig 2). In both control and sIBM-derived myotubes, the expression of human muscle stem cell markers such as Integrin alpha 7 (ITGA7), m-cadherin (CDH15), and CD82 was observed to be high (S1 Fig). The results revealed a total of 104 genes that exhibited a significant upregulation of more than two-fold in sIBM (Table 2).

Among these upregulated genes, myoblast differentiation factor, myogenin was upregulated in sIBM, as reported previously [30]. Conversely, thirteen differentially expressed genes (DEGs) were downregulated by less than twofold (Table 3).

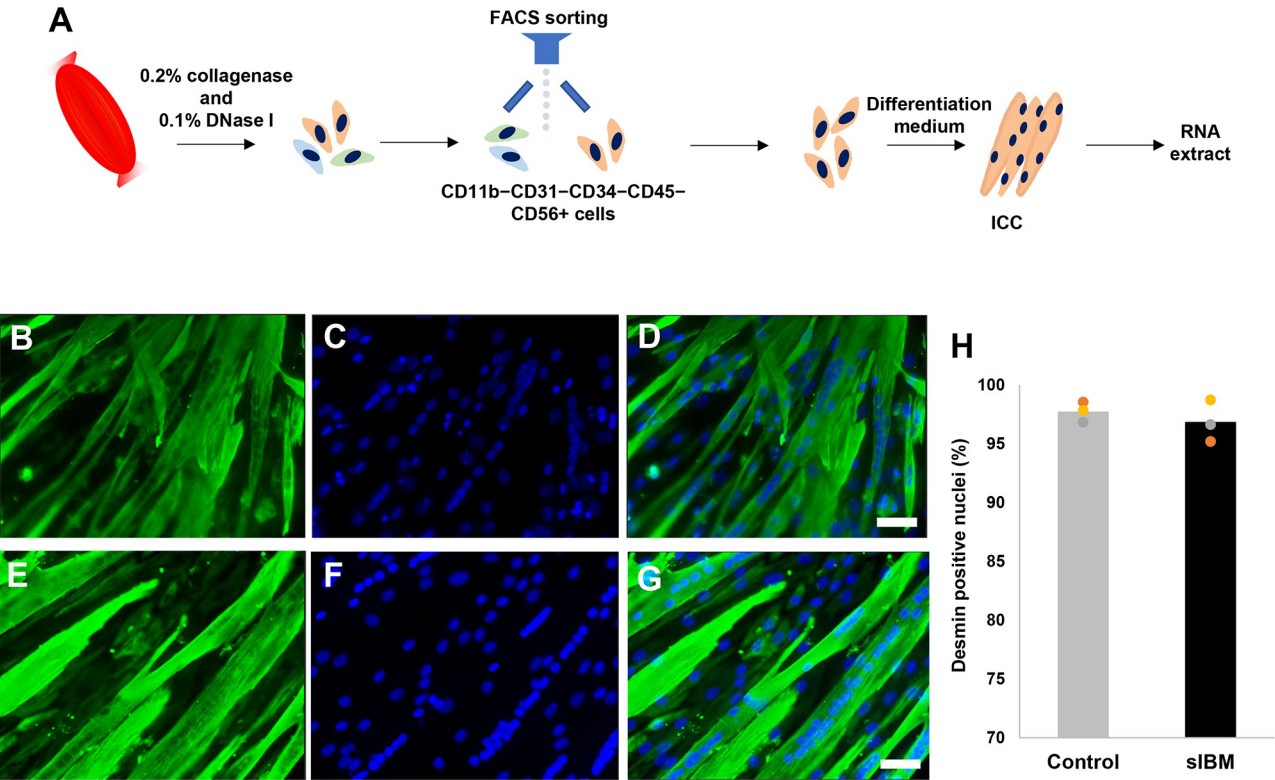

**Fig 1. Myotubes derived from patient muscle biopsy samples.** A. Schema of collecting myoblast from the muscle biopsy sample. The tissue was minced and digested with 0.2% collagenase and 0.1% DNase I. CD11b⁻CD31⁻ CD34⁻ CD45⁻ CD56⁺ cells were sorted using FACS. Purified myoblasts were expanded in the growth medium. Myoblasts were differentiated into myotubes in the differentiation medium. B–G. Myotubes derived from patients with sIBM (B, C) and control (D, E). Desmin (B, D) and Dapi (C, E) staining. Overlay images are also shown (D, G). Scale bar, 50 μm. H. 95–98% of the nuclei exhibited colocalization with Desmin. sIBM, sporadic inclusion body myositis.

In order to encompass a broader spectrum of genes, those with a 1.5-fold or greater change were systematically cataloged, resulting in a total of 991 altered genes (S2 Table). Visual representation through heat maps effectively segregated sIBM and control myotubes, confirming the expected distinctions (Fig 2A). Gene Ontology (GO) term classification disclosed enrichment in biological processes (GO-BP) associated with skeletal muscle function, particularly in pathways such as muscle filament sliding and muscle contraction. Molecular function (GO-MF) analysis highlighted enrichment in actin filament binding and structural constituents of muscle (Fig 2B and 2C), reaffirming the specificity of changes linked to skeletal muscle cells.

To assess the dissimilarities in expression profiles between myotubes and muscle biopsies, RNA-seq analysis was performed on biopsy samples from individuals with sIBM (n = 3) and disease control (n = 4) samples (referred to as "biopsy" samples: S2 Table). A total of 901 genes exhibited more than a 1.5-fold variation in muscle biopsy samples. Notably, proteasome (prosome and macropain) subunit, beta type, 8 (PSMB8), a member of the interferon-2 pathway, was increased, consistent with previous analyses of biopsied muscle [31]. Other molecules in the interferon-2 pathway, including guanylate binding protein 1, interferon-inducible (GBP1), and GBP2 [31], were also elevated, along with upregulation of CCL13, interferon regulatory factor 8 (IRF8), CCR5, VCAM1, HLA-DRA, TYROBP, complement component 1, q subcomponent, B chain (C1QB), major histocompatibility complex class II, DR beta 1 (HLA-DRB1), CD74, and CXCL9 [8, 13, 14].

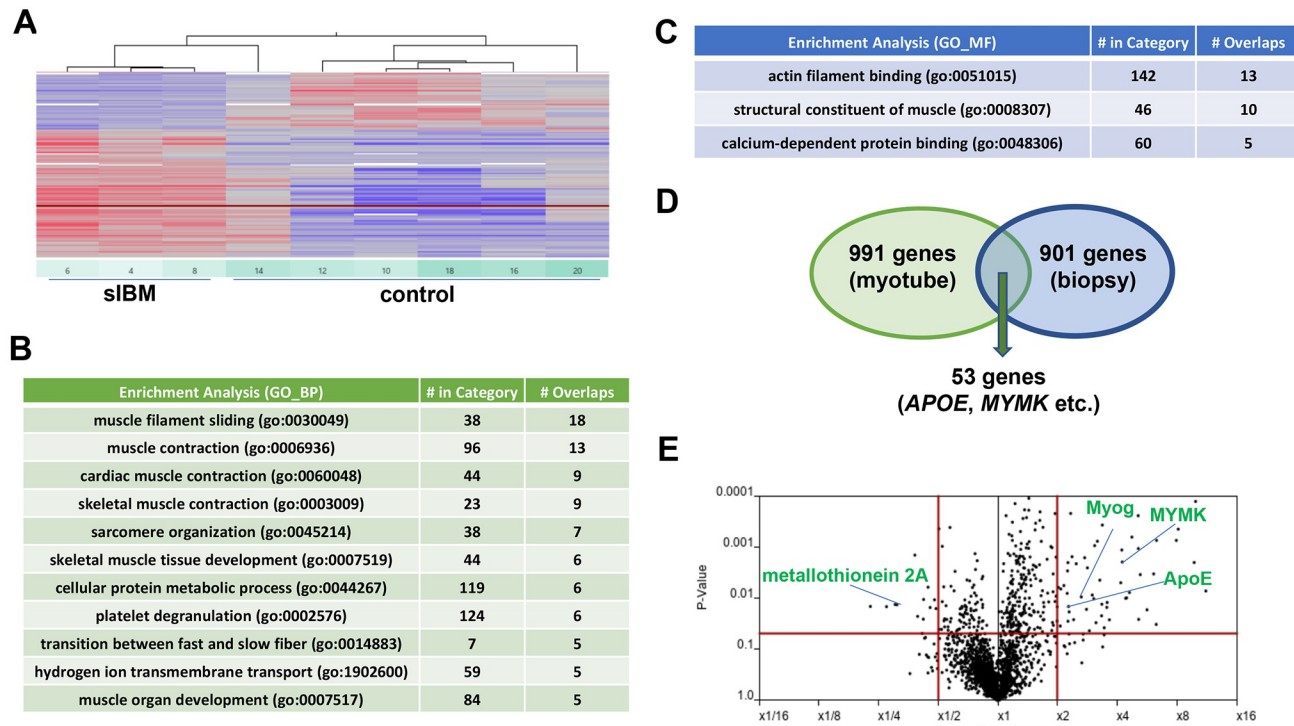

**Fig 2. Intrinsic cell-autonomous characterization of myotubes derived from sIBM.** A. Heat maps of three sIBM and six control myotube samples. B. The enrichment of the genes differentially expressed in the sIBM and control groups was analyzed by GO_BP term analysis using the Subio platform. The top terms are listed with >4 overlapped genes, with p-values < 0.05 (Fisher's exact test). C. The enrichment of the genes differentially expressed in the sIBM and control groups were analyzed by GO_MF term analysis using the Subio platform. The top terms are listed with >4 overlapped genes, with p-values < 0.05 (Fisher's exact test). D. Venn diagram showing overlapped dysregulated genes between sIBM myotubes (991 genes) and sIBM biopsy (901 genes). Differentially expressed genes (DEGs: fold-change difference of |1.5|) were compared. Fifty-three genes are commonly dysregulated. E. Volcano plot of the genes differentially expressed in sIBM and SSC focusing genes are plotted on the figures. sIBM, sporadic inclusion body myositis.

Then, a comparative analysis was conducted to determine the extent of the difference between the muscle biopsy and myotube samples (Fig 2D, S3 Table). Of the 53 extracted genes, 32 (60%) exhibited opposing directions of change in expression (e.g., increased in biopsy vs decreased in myotube). Among the commonly upregulated genes, transmembrane protein 8C (TMEM8C; also known as a myomaker, MYMK), myogenin (Myog), and ApoE were found, whereas metallothionein 2A (MT2A) was found among the downregulated genes (Fig 2E).

Among genes that were commonly altered in muscle biopsies and myotubes, molecules related to muscle differentiation and degeneration were examined at the protein level. MYMK is a myoblast fusion-associated molecule [32]. Deletion from myoblasts was reported to exacerbate symptoms in mouse models of muscular dystrophy [33]. The expression of MYMK was high in both myotubes and biopsy samples (Fig 3A and 3B). In immunostaining, the number of MYMK-positive myoblasts was higher in sIBM samples than in the control samples (Fig 3C).

The investigation extended to ApoE, known for its abnormal accumulation in sIBM and its potential role in muscle degenerative pathology [18–20]. High expression of ApoE in sIBM was found in the RNA-seq analysis of both myoblasts and muscle biopsies (Fig 4A and 4B). Immunostaining also showed high expression of ApoE in sIBM (Fig 4C), as shown previously [18–20]. Moreover, staining for amyloid oligomer A11 was enhanced in sIBM (Fig 4D).

**Table 2. Upregulated genes in sIBM myotube (vs control, >×2, p < 0.05, Student's t-test (equal variance)).** Cell-autonomous expression analysis from sIBM myotube. A total of 104 genes are listed. sIBM, sporadic inclusion body myositis.

| Gene | Gene Name | Fold Change | P-Value |
|---|---|---|---|
| MYL1 | myosin light chain 1 | 11.202059 | 0.007286711 |
| MYL2 | myosin | 9.947378 | 1.29E-04 |
| TNNT3 | troponin T3, fast skeletal type | 9.788654 | 0.002023014 |
| ACTA1 | actin, alpha 1, skeletal muscle | 8.129974 | 4.51E-04 |
| MYBPH | myosin binding protein H | 7.952435 | 7.49E-04 |
| TNNC2 | troponin C2, fast skeletal type | 6.319282 | 7.47E-04 |
| F13A1 | coagulation factor XIII A chain | 6.295651 | 0.03281219 |
| MYLPF | myosin light chain, phosphorylatable, fast skeletal muscle | 6.110357 | 0.003378833 |
| COX6A2 | cytochrome c oxidase subunit 6A2 | 5.6563983 | 0.017002339 |
| TNNT1 | troponin T1, slow skeletal type | 5.2325187 | 0.003490027 |
| MYH3 | myosin heavy chain 3 | 5.1348267 | 0.026199384 |
| TNNC1 | troponin C1, slow skeletal and cardiac type | 5.116753 | 0.00108173 |
| MYL6B | myosin light chain 6B | 5.106448 | 2.46E-04 |
| HSPB8 | heat shock protein family B (small) member 8 | 4.745588 | 6.45E-04 |
| TNNI1 | troponin I1, slow skeletal type | 4.6149263 | 0.007806895 |
| MYL4 | myosin light chain 4 | 4.465095 | 0.009941747 |
| DES | desmin | 4.4114947 | 0.010139073 |
| BIN1 | bridging integrator 1 | 4.243511 | 0.001979603 |
| ENO3 | enolase 3 | 4.2209435 | 0.001152897 |
| TMEM8C | transmembrane protein 8C | 4.1701674 | 0.005040719 |
| KLHL41 | kelch like family member 41 | 3.6543784 | 0.026659327 |
| SLN | sarcolipin | 3.5593212 | 0.012243188 |
| STAC3 | SH3 and cysteine rich domain 3 | 3.5460794 | 0.015242067 |
| ANKRD1 | ankyrin repeat domain 1 | 3.441077 | 0.038272817 |
| ACTN2 | actinin alpha 2 | 3.3681853 | 3.75E-04 |
| MIR133B | microRNA 133b | 3.3491333 | 0.013239365 |
| MYOG | myogenin | 3.3376992 | 0.022312023 |
| CASQ2 | calsequestrin 2 | 3.2697413 | 0.004478621 |
| TPM3 | tropomyosin 3 | 3.1789114 | 7.78E-04 |
| IL32 | interleukin 32 | 3.1274703 | 0.030406158 |
| MYH7 | myosin heavy chain 7 | 3.0794365 | 9.46E-04 |
| MEF2C | myocyte enhancer factor 2C | 3.047847 | 0.010303746 |
| FABP3 | fatty acid binding protein 3 | 2.9947274 | 0.008757984 |
| ACTC1 | actin, alpha, cardiac muscle 1 | 2.895736 | 0.043732658 |
| HSPB3 | heat shock protein family B (small) member 3 | 2.8300886 | 0.002575204 |
| RASSF4 | Ras association domain family member 4 | 2.8004258 | 0.016659606 |
| MURC | - | 2.7417965 | 0.001631513 |
| HN1 | - | 2.6370249 | 0.009545088 |
| CDKN1A | cyclin dependent kinase inhibitor 1A | 2.485724 | 0.002141749 |
| DMPK | dystrophia myotonica protein kinase | 2.4113045 | 0.02649946 |
| MLLT11 | myeloid/lymphoid or mixed-lineage leukemia; translocated to, 11 | 2.3682325 | 0.00156106 |
| CFL2 | cofilin 2 | 2.3300521 | 2.23E-04 |
| UCP2 | uncoupling protein 2 | 2.2923715 | 6.50E-05 |
| CKM | creatine kinase, M-type | 2.2887108 | 0.018692916 |
| DCLK1 | doublecortin like kinase 1 | 2.27428 | 0.014638593 |
| APOE | apolipoprotein E | 2.258964 | 0.02849325 |

*(Continued)*

**Table 2.** (Continued)

| Gene | Gene Name | Fold Change | P-Value |
|---|---|---|---|
| STMN1 | stathmin 1 | 2.2580466 | 0.002031321 |
| SNORD24 | - | 2.2476733 | 0.002950982 |
| SETD7 | SET domain containing lysine methyltransferase 7 | 2.2390144 | 0.00811696 |
| ZNF106 | zinc finger protein 106 | 2.1539176 | 0.005711007 |
| PRDX6 | peroxiredoxin 6 | 2.078649 | 0.006655635 |
| MRAS | muscle RAS oncogene homolog | 2.0655692 | 0.035309568 |
| MIR4787 | microRNA 4787 | 2.021805 | 0.006221586 |
| CYB5R1 | cytochrome b5 reductase 1 | 1.9932206 | 0.014857224 |
| NEXN | nexilin F-actin binding protein | 1.9889293 | 0.028174449 |
| BLCAP | bladder cancer associated protein | 1.9670132 | 0.009212402 |
| MYH8 | myosin heavy chain 8 | 1.965625 | 0.001588375 |
| MAPRE3 | microtubule associated protein RP/EB family member 3 | 1.9434834 | 2.23E-04 |
| TUBA1A | tubulin alpha 1a | 1.9425162 | 0.011722326 |
| KIF5B | kinesin family member 5B | 1.9053978 | 0.03412914 |
| ATP2A2 | ATPase sarcoplasmic/endoplasmic reticulum Ca2+ transporting 2 | 1.9013083 | 0.003384043 |
| MIR27A | - | 1.8992655 | 0.007567231 |
| CTSH | cathepsin H | 1.8868161 | 0.030131867 |
| SMPX | small muscle protein, X-linked | 1.876876 | 0.001038277 |
| TPM2 | tropomyosin 2 (beta) | 1.8715552 | 0.043131925 |
| FKBP3 | FK506 binding protein 3 | 1.8516325 | 9.88E-04 |
| KLHDC2 | kelch domain containing 2 | 1.8059789 | 0.011938629 |
| HNRNPDL | heterogeneous nuclear ribonucleoprotein D like | 1.7927606 | 7.21E-05 |
| RTN2 | reticulon 2 | 1.7837111 | 0.008038428 |
| UQCR10 | ubiquinol-cytochrome c reductase, complex III subunit X | 1.7718846 | 0.001749373 |
| IDH2 | isocitrate dehydrogenase (NADP(+)) 2, mitochondrial | 1.7091504 | 0.014825196 |
| ODC1 | ornithine decarboxylase 1 | 1.7080046 | 0.026313692 |
| HSPB2 | heat shock protein family B (small) member 2 | 1.7018151 | 0.012277631 |
| RAPSN | receptor associated protein of the synapse | 1.6918406 | 0.01916177 |
| ITGA6 | integrin subunit alpha 6 | 1.6912874 | 0.026712555 |
| MIR3960 | - | 1.6898465 | 4.16E-04 |
| SYNPO2 | synaptopodin 2 | 1.6888578 | 5.24E-05 |
| SBDS | SBDS, ribosome assembly guanine nucleotide exchange factor | 1.6792147 | 0.007196609 |
| PRUNE2 | prune homolog 2 | 1.6792002 | 0.04550971 |
| TRIM72 | tripartite motif containing 72 | 1.6562946 | 7.82E-04 |
| WSB2 | WD repeat and SOCS box containing 2 | 1.6523238 | 0.009012659 |
| CHCHD10 | coiled-coil-helix-coiled-coil-helix domain containing 10 | 1.6480478 | 0.004553978 |
| AK1 | adenylate kinase 1 | 1.6430076 | 0.022719005 |
| NDUFS5 | NADH:ubiquinone oxidoreductase subunit S5 | 1.6388379 | 0.004627282 |
| CALM1 | calmodulin 1 | 1.6350479 | 0.03881275 |
| GYS1 | glycogen synthase 1 | 1.6332341 | 2.07E-04 |
| SYTL2 | synaptotagmin like 2 | 1.628396 | 3.04E-04 |
| TIMM8B | translocase of inner mitochondrial membrane 8 homolog B | 1.6226355 | 0.002241131 |
| HACD1 | 3-hydroxyacyl-CoA dehydratase 1 | 1.6016754 | 8.34E-05 |
| CACNB1 | calcium voltage-gated channel auxiliary subunit beta 1 | 1.5958093 | 0.01881004 |
| FNDC5 | fibronectin type III domain containing 5 | 1.5868237 | 0.037550054 |
| RAPGEF1 | Rap guanine nucleotide exchange factor 1 | 1.5625403 | 0.002484248 |
| CCT2 | chaperonin containing TCP1 subunit 2 | 1.554814 | 0.004175162 |

*(Continued)*

**Table 2.** (Continued)

| Gene | Gene Name | Fold Change | P-Value |
|---|---|---|---|
| SYNC | syncoilin, intermediate filament protein | 1.5515491 | 3.46E-04 |
| COX6C | cytochrome c oxidase subunit 6C | 1.5454249 | 0.00239981 |
| HRC | histidine rich calcium binding protein | 1.5410172 | 4.67E-04 |
| COX11 | COX11, cytochrome c oxidase copper chaperone | 1.5325474 | 1.85E-05 |
| GRASP | general receptor for phosphoinositides 1 associated scaffold protein | 1.5297244 | 0.040888462 |
| NDUFS2 | NADH:ubiquinone oxidoreductase core subunit S2 | 1.5288111 | 5.10E-05 |
| UQCRC2 | ubiquinol-cytochrome c reductase core protein II | 1.5287356 | 0.003558046 |
| SRPK3 | SRSF protein kinase 3 | 1.5262501 | 0.01884095 |
| UQCRQ | ubiquinol-cytochrome c reductase complex III subunit VII | 1.5190833 | 0.018657938 |
| MAP1A | microtubule associated protein 1A | 1.5157436 | 0.017696884 |
| GLO1 | glyoxalase I | 1.5097214 | 0.002272782 |

The study then turned its attention to the quantification of ApoE proteins in myotubes. As anticipated, ApoE proteins were significantly increased in sIBM compared to control myotubes (Fig 5A and 5B, S1 Raw images). In addition, myogenin proteins were significantly increased in sIBM myotubes (Fig 5A and 5C).

## Discussion

In this study, myotubes derived from sIBM-associated myoblasts were established, and a comprehensive RNA-seq analysis was conducted to delineate gene expression profiles specific to myotubes (Tables 2 and 3). By cross-referencing the RNA-seq results between the "myotube" and "biopsy" muscle, we could identify MYMK and ApoE, implicated in the myofiber-intrinsic contribution to the pathogenesis of sIBM (Fig 2, S3 Table).

A noteworthy observation was the inverse regulation of many genes upregulated in muscle biopsy, predominantly inflammatory genes, which exhibited significant downregulation in sIBM myotubes compared to control myotubes (S3 Table). Only 21 genes displayed shared directional changes in expression in both muscle biopsy and myotube samples, indicating a

**Table 3. Downregulated genes in sIBM myotube (vs SSC, >×2, p < 0.05, Student's t-test (equal variance)).** Cell-autonomous expression analysis from sIBM myotube. Thirteen genes are listed. sIBM, sporadic inclusion body myositis.

| Gene | Gene Name | Fold Change | P-Value |
|---|---|---|---|
| SERPINE2 | serpin family E member 2 | 0.22732568 | 0.014635123 |
| GAS6 | growth arrest specific 6 | 0.27426982 | 0.014784024 |
| SNORD50B | - | 0.3030194 | 0.013391182 |
| SNORD30 | - | 0.30977944 | 0.013535343 |
| IGFBP7 | insulin like growth factor binding protein 7 | 0.3813738 | 0.001449735 |
| MT2A | metallothionein 2A | 0.3937795 | 0.022868155 |
| IGFBP4 | insulin like growth factor binding protein 4 | 0.41521716 | 0.049309194 |
| MIR1282 | microRNA 1282 | 0.42054167 | 0.025922615 |
| IGFBP5 | insulin like growth factor binding protein 5 | 0.42213562 | 0.010698475 |
| ITGBL1 | integrin subunit beta like 1 | 0.44444004 | 0.006151323 |
| ITGB5 | integrin subunit beta 5 | 0.45522952 | 0.015339294 |
| CTHRC1 | collagen triple helix repeat containing 1 | 0.47890285 | 0.01084614 |
| MXRA8 | matrix remodeling associated 8 | 0.4874133 | 0.016108675 |

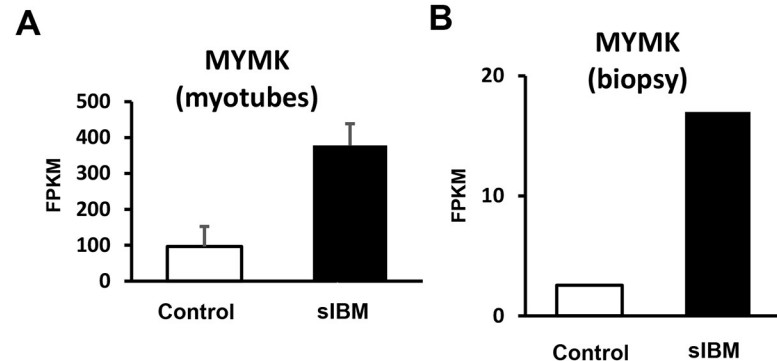

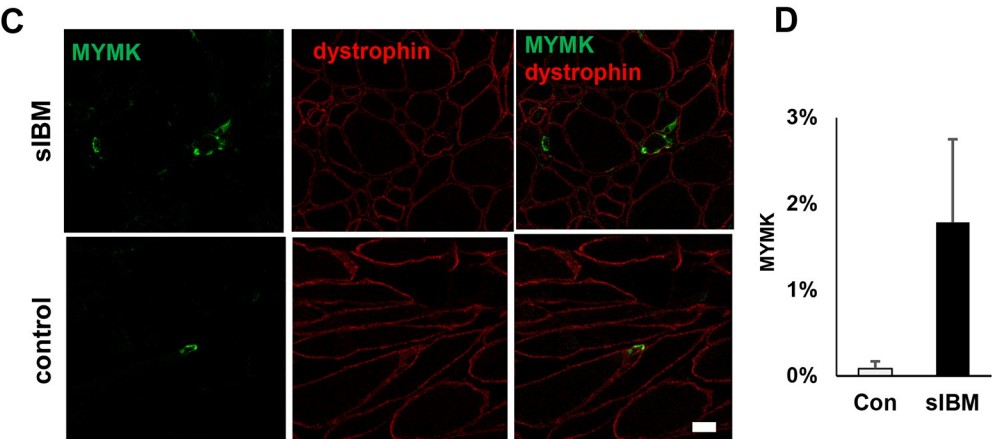

**Fig 3. Expression of MYMK in both myotubes and sIBM biopsy sections.** A. MYMK mRNA upregulation in sIBM myotubes. B. MYMK mRNA is upregulated in the sIBM biopsy muscle. (p-value = 0.00005, q-value = 0.002035). C. MYMK is found in the sIBM biopsy section. Scale bar, 10 μm. D. Quantification of the MYMK positive fibers (%).

potential limitation in relying solely on myotubes to encapsulate the intricate pathology of sIBM (S3 Table).

Significantly, members of the interferon-2 pathway, such as PSMB8, GBP1, and GBP2, were elevated in biopsy samples, aligning with prior analyses of biopsied muscle [31]. This change is not observed in myotubes. This activation might represent a non-cell-autonomous mechanism contributing to muscle degeneration in sIBM. It may simply be expressed at low levels in myotubes but not reaching the threshold for detection, which is a limitation of our analysis in this study. Discriminating between cell types, such as through single-cell RNA-seq analysis, is a task for future research.

Given that changes in skeletal muscle- and myoblast-related molecules such as ApoE and myogenin (Figs 2, 4, and 5), which have been repeatedly noted in biopsies of sIBM previously [18–20, 30], were successfully identified in myotube and biopsied muscle, our analysis may be able to capture intrinsic changes in skeletal muscles in sIBM.

ApoE and lipoprotein receptors are also found in rimmed vacuole structure, suggesting an aberrant cholesterol metabolism in sIBM muscles [19, 20]. ApoE stimulates amyloid precursor protein transcription and amyloid beta secretion robustly in human neurons [34]. Amyloid beta-overexpressed mouse was used as the sIBM model [21]. The clarification of the

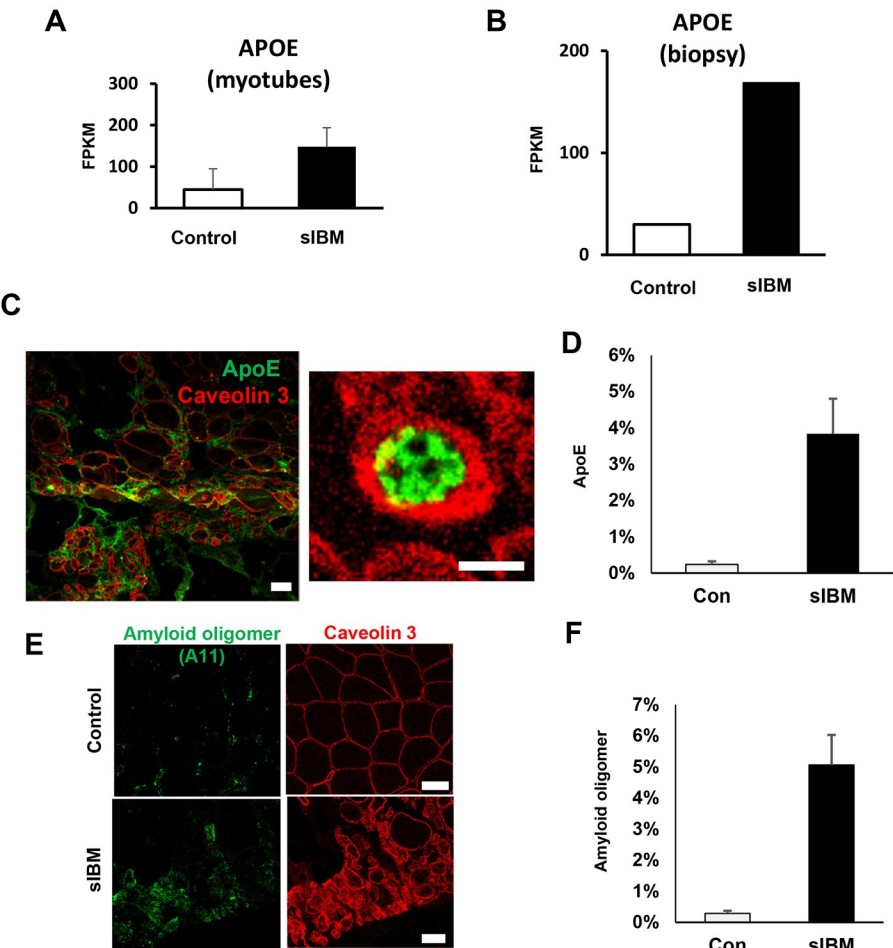

**Fig 4. Expression of ApoE in sIBM biopsy sections.** A. ApoE mRNA upregulation in sIBM myotubes. B. ApoE mRNA is upregulated in the sIBM biopsy muscle. (p-value = 0.00005, q-value = 0.002035). C. ApoE protein is upregulated in the sIBM biopsy section. Scale bar, 10 μm. D. Quantification of the ApoE positive fibers (%). E. Amyloid oligomer (A11) is upregulated in the sIBM biopsy section. Scale bar, 50 μm. F. Quantification of the Amyloid oligomer positive fibers (%).

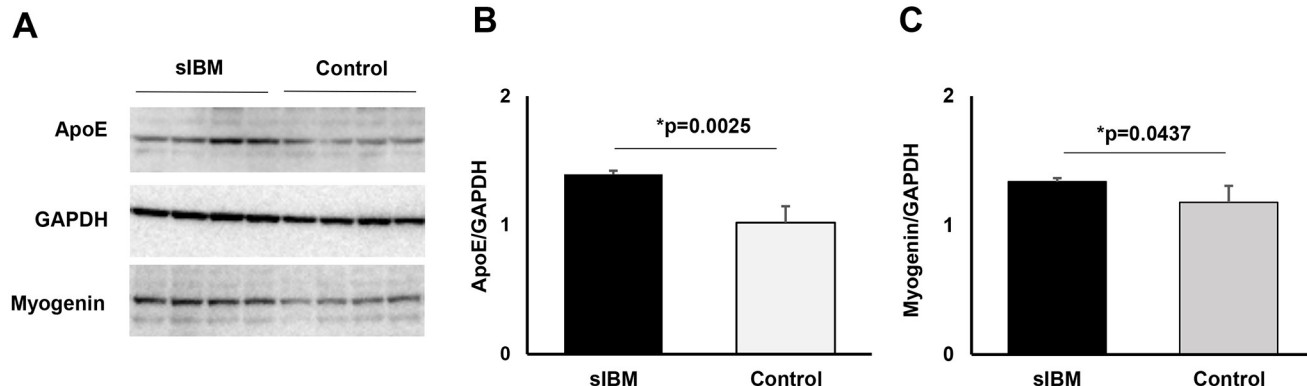

**Fig 5. Protein levels of ApoE and myogenin in sIBM myotubes.** A. Representative blot of ApoE, GAPDH, and myogenin in sIBM and control myotubes. B. The protein level of ApoE is significantly increased in sIBM. P = 0.0025. Students' t-test. C. The protein level of myogenin is significantly increased in sIBM. P = 0.0437. Students' t-test.

mechanism by which ApoE is high in sIBM myotubes may lead to the control of ApoE-mediated abnormal protein accumulation and muscle degeneration. Additionally, it is possible that Amyloid beta overexpression might influence the characteristic KLRG1-T cells of inclusion body myositis [35] via the insulin-like growth factor signaling pathway [36] or immune modulation mechanisms [37]. Previously, we showed that the RNA-binding protein TDP-43 was localized from the nucleus to the cytoplasm in electric pulse stimulation culture [38]. Similar to ApoE, TDP-43 accumulates in the skeletal muscles in sIBM [39]. Patient-derived myotube cells may be useful for the study of the regulation of neurodegenerative disease-related proteins.

MYMK, a myomaker, is a well-conserved plasma membrane protein required for myoblast fusion to form multinucleated myotubes [32, 40]. Myogenin is also upregulated in sIBM myotubes [30]. Myogenin binds to the MYMK promoter and is required for the expression of MYMK and other genes essential for myocyte fusion [41]. Myogenin is also upregulated in sIBM muscle biopsy [30]. In denervation-induced muscle atrophy, myogenin plays both a regulator of muscle development and an inducer of neurogenic atrophy [42]. Given that enrichment analysis also captured changes in muscle contraction and development (Fig 2B and 2C), the triggering of muscle atrophy signaling and muscle differentiation via MYMK or myogenin may be involved in the pathogenesis of sIBM.

This study has several limitations. First, the cell culture condition might have changed the intrinsic gene expression in sIBM myotubes. Single-nuclei RNA-seq using human muscle biopsy could solve the intrinsic gene expression in myotubes *in vivo*. Second, SSC was defined as the control, although it was derived from a pathological joint in the RCT and therefore did not have normal function. In this study, we did not use inflammatory muscle disorders with an onset at a younger age, such as polymyositis, as controls in order to match the age range more effectively. Third, the quantity of sorted myoblasts was contingent on several factors, including the expression of antibody markers, the specific anatomical region of the muscle subjected to biopsy, and the uniformity and extent of digestion facilitated by collagenase. Lastly, the study was constrained by a limited number of samples.

In conclusion, the study elucidates the cell-autonomous profiles of sIBM-derived myotubes. ApoE and MYMK are commonly upregulated in both sIBM-derived myotubes and biopsy samples. This comprehensive catalog of gene expression changes stands as a valuable resource for future investigations into the pathogenesis of sIBM, with a specific focus on primary muscle degeneration.

## Supporting information

**S1 Fig. The expression of human muscle stem cell markers.**
(TIF)

**S1 Raw images. Protein levels of ApoE, myogenin and GAPDH in sIBM and control myotubes.**
(PDF)

**S1 Table. Unmapped reads.**
(XLSX)

**S2 Table. Dysregulated genes in sIBM myotube (vs SSC, >×1.5).** A total of 991 genes are listed.
(XLSX)

**S3 Table. Dysregulated genes in sIBM biopsy samples (vs disease control, >×1.5).** A total of 901 genes are listed.
(XLSX)

**S4 Table. Common dysregulated genes in sIBM myotube and biopsy samples.** A total of 53 genes are listed. Although genes are listed, most of the upregulated genes in muscle biopsy samples are downregulated in myotubes.
(XLSX)

## Acknowledgments

We thank Naoko Shimakura, Akiko Machii, Mai Kakinuma, and Hinako Shigihara (Tohoku University, Japan) for general technical support and Enago for the English language review (www.enago.jp). We also thank Ayami Otsuki for the cell culture and Dr. Maki Tateyama (National Hospital Organization Iwate Hospital, Japan) for useful technical advice and discussions.

## Author Contributions

**Conceptualization:** Naoki Suzuki, Masashi Aoki.

**Data curation:** Naoki Suzuki, Rumiko Izumi, Masako Suzuki, Akiyuki Ohno, Hiroya Ono, Kensuke Ikeda.

**Formal analysis:** Naoki Suzuki, Masako Suzuki.

**Funding acquisition:** Takaaki Abe, Ryoichi Nagatomi, Masashi Aoki.

**Investigation:** Naoki Suzuki, Makoto Kanzaki, Masashi Koide, Rumiko Izumi, Ryo Fujita, Tadahisa Takahashi, Kazumi Ogawa, Yutaka Yabe, Masahiro Tsuchiya, Masako Suzuki, Ryuhei Harada, Akiyuki Ohno, Hiroya Ono, Naoko Nakamura, Kensuke Ikeda, Hitoshi Warita, Shion Osana, Yoshitsugu Oikawa, Takafumi Toyohara, Takaaki Abe, Muliang Rui, Ryoichi Nagatomi, Yoshihiro Hagiwara, Masashi Aoki.

**Resources:** Naoki Suzuki, Makoto Kanzaki, Masashi Koide, Rumiko Izumi, Ryo Fujita, Tadahisa Takahashi, Kazumi Ogawa, Masako Suzuki, Yoshihiro Hagiwara.

**Validation:** Muliang Rui, Satoru Ebihara.

**Visualization:** Rumiko Izumi.

**Writing – original draft:** Naoki Suzuki.

**Writing – review & editing:** Naoki Suzuki, Ryuhei Harada, Masashi Aoki.

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
