## [Decision Letter · Decision Letter 0]

16 Apr 2024

PONE-D-24-02730Sporadic inclusion body myositis-derived myotube culture revealed muscle cell-autonomous expression profilesPLOS ONE

Dear Dr. Suzuki,

Thank you for submitting your manuscript to PLOS ONE. After careful consideration, we feel that it has merit but does not fully meet PLOS ONE’s publication criteria as it currently stands. Therefore, we invite you to submit a revised version of the manuscript that addresses the points raised during the review process.

**Two reviewers have reviewed and expressed potential interest. Please work to address all concerns raised by the reviewers prior to publication.**

We look forward to receiving your revised manuscript.

Kind regards,

Keisuke Hitachi

Academic Editor

PLOS ONE

Journal Requirements:

This research was partially supported by Intramural Research Grants 29-4 and 2-5 for Neurological and Psychiatric Disorders provided to M.A. from the National Center of Neurology and Psychiatry of Japan; the Practical Research Project for Rare/Diseases (20dk0310086) provided to M.A. and Moonshot R&D Program (JPMJMS 23zf0127001h0003) to T.A. from the Japan Agency for Medical Research and Development (AMED); Grants-in-Aid for Research on Rare and Intractable Diseases (H29-nanchitou(nan)-ippan-030 and 20CF1036) provided to M.A. from the Ministry of Health, Labor and Welfare of Japan; a Grant-in-Aid for Challenging Exploratory Research (20K21563) provided to M.A. and N.S., Scientific Research C (18K07519) provided to N.S., from the Japanese Ministry of Education, Culture, Sports, Science and Technology. This research was also supported by the Cooperative Research Project Program of the Joint Usage/Research Center at the Institute of Development, Aging and Cancer, Tohoku University.

4. Thank you for uploading your study's underlying data set. Unfortunately, the repository you have noted in your Data Availability statement does not qualify as an acceptable data repository according to PLOS's standards.

5. Please include your tables as part of your main manuscript and remove the individual files. Please note that supplementary tables (should remain/ be uploaded) as separate "supporting information" files.

7. PLOS ONE now requires that authors provide the original uncropped and unadjusted images underlying all blot or gel results reported in a submission’s figures or Supporting Information files. This policy and the journal’s other requirements for blot/gel reporting and figure preparation are described in detail at https://journals.plos.org/plosone/s/figures#loc-blot-and-gel-reporting-requirements and https://journals.plos.org/plosone/s/figures#loc-preparing-figures-from-image-files. When you submit your revised manuscript, please ensure that your figures adhere fully to these guidelines and provide the original underlying images for all blot or gel data reported in your submission. See the following link for instructions on providing the original image data: https://journals.plos.org/plosone/s/figures#loc-original-images-for-blots-and-gels. 

Reviewers' comments:

Reviewer's Responses to Questions

**Comments to the Author**

1. Is the manuscript technically sound, and do the data support the conclusions?

Reviewer #1: Partly

Reviewer #2: Partly

2. Has the statistical analysis been performed appropriately and rigorously? 

Reviewer #1: I Don't Know

Reviewer #2: Yes

3. Have the authors made all data underlying the findings in their manuscript fully available?

Reviewer #1: Yes

Reviewer #2: Yes

4. Is the manuscript presented in an intelligible fashion and written in standard English?

Reviewer #1: Yes

Reviewer #2: Yes

5. Review Comments to the Author

**Reviewer #1:** A straightforward study examining the similarities between myotubes derived from myoblasts from inclusion body myositis patients and age matched controls. These samples are assessed by RNA-seq and compared to RNA-seq data from whole muscle biopsies. Overall, the study is interesting but several areas would strengthen the conclusions and allows for more complete interpretation of the data.

Methods: Central to this study is the isolation of myogenic precursor cells from human muscle biopsies. The protocol used (FACS sorting) relies heavily on negative sorts to isolate myoblasts (MuSCs?) and only CD56 is used as a positive sort. To increase confidence, the isolated cells should also be subjected to a positive sort, or in this case, since the RNA-seq analysis is complete, that validation of the purity of the cells used for differentiation be used with accepted markers for human MuScs (eg. Integrin alpha 7, m-cadherin, CD82).

Figure 1B-E. Overlay of the DAPI and desmin stain is critical to appreciate the % of cells that have not differentiated. While this appears to be high for both conditions, a calculation of the differentiation index would also help appreciate the level of contamination from reserve cells and the expected variability from sample to sample and experiment to experiment. This has an impact on the downstream analysis and the interpretation of the findings.

Figure 2. For the RNA-seq analysis, it would be beneficial to include the FDR cut-offs used for significance in the text. Changes in the expression of factors associated with differentiation and myofiber maturity/contraction can be related to differences in the efficiency of differentiation between samples.

For the comparative analysis, it is not surprising that genes that are uniquely expressed in myofibers are also commonly regulated in whole muscle samples. More interesting however, is the massive number of genes oppositely regulated. No further analysis is provided here, but would be very useful in terms of understanding the origins. Was a similar analysis done for controls (or were the DEGs determined as relative to controls?). It would be interesting to see how divergent the control myotubes are from the healthy whole muscle, as a way to infer differences that the microenvironment contributes.

For the comparison between myotubes generated in culture and whole muscle biopsy samples, The gene name for Tmem8c is actually MYMK and this should be changed throughout.

Figure 3. The text says that the levels were verified at the protein level, however FPKM are shown and no quantification over the patient samples are provided for the immunostained images. Quantifications are essential to validate the conclusions.

Figure 4. Similar comments to Figure 3.

Discussion:

Line 323. I do not think the word pathogenesis applied to the myotubes in culture. I appreciate that you mean a myofiber-intrinsic contribution to the pathogenesis of the disease, but this is not clear as written.

This statement (lines 332-336) “The examination of muscle biopsies suggested altered gene expression in lymphocytes and other immunopathological conditions, phenomena not mirrored in myotube samples.” Is not supported by the data shown.

The statement in lines 337-340 is also unclear as to where the findings come from (as related to SASP). Please clarify.

Minor comments: Care should be taken to use the appropriate gene nomenclature for humans throughout to avoid confusion with mouse genes or protein names.

**Reviewer #2:** Suzuki et al. present an interesting finding regarding the pathological mechanism(s) involved in sIBM. Previous findings have failed to define the pathology as autoimmune or degenerative, not having determined whether the pathology observed is primarily due to intrinsic or extrinsic forces. In their study the authors confront the gene expression profile using RNA-seq between sIBM biopsy samples and controls as well as with myotubes derived from myoblasts isolated from control and sIBM biopsies. In such a manner, the study is able to differentiate between gene expression profiles of sIBM biopsies (due both to intrinsic and extrinsic forces), and gene expression profiles of in vitro differentiated sIBM-derived myotubes (due to intrinsic forces). The data are interesting and find similar results to previous reports. What is missing is a full consideration of the findings.

Major comments:

1. Figure 1, please show the merged image of DAPI and desmin.

2. As sIBM leads to muscle death and atrophy, it is important to known if any difference was observed in the differentiation potential/rate and viability between controls and sIBM biopsy-derived myoblasts.

3. Data from myotubes suggests a potential alteration of the insulin growth factor response and cellular adhesion in sIBM. How does this fit in with previous findings in the field?

4. In discussing differences between data in Supplementary Table 2 and Supplementary Table 3, the authors observe the absence of immunopathological gene expression (lines 333-341) in myotubes and suggest that some phenotypic findings may be linked to STING and non-cell-autonomous mechanisms. The authors should be careful here with this interpretation. First, there is more than STING to consider; second, data in Supplemental Table 2 could suggest a defective type I interferon response; and third, the authors used a cut-off for choosing which genes to include in their analysis, but it should be noted that in almost all cases, the ratio of gene expression of common genes between sIBM-derived myotubes/control versus sIGM biopsy/control is always decreased. therefore some changes seen in the biopsy samples may have been below the threshold for consideration in the myotubes, but none-the-less still present.

5. The authors focus then on ApoE, TMEM8C and myogenin. The pathophysiological role that upregulation of these proteins has in sIBM is difficult to determine. TMEM8C and myogenin should, at face value, favor muscle development. Likewise, the role of ApoE is difficult to determine due to its varying immunomodulatory roles, including T cell suppression.

6. From the RNA-seq data, the authors should state the number of reads that could not be matched for each experiment. Additionally, in regard to the 53 genes common to myotubes and biopsies, the authors should look at the sequence and exon usage to determine if any of these common genes contain mutations or are potentially alternatively spliced between control and sIBM, including ApoE and TMEM8C (use of a monoclonal antibody may not necessarily catch all isoforms).

Minor comments:

1. Please define the supplier DBS, line 221.

2. For the lysis buffer on line 233, add "(RIPA)" between "radio-immunoprecipitation" and "buffer".

3. In Supplemental Table 3 it might be beneficial to organize the table as those genes that commonly increase in sIBM versus those that increase in biopsy but decrease in myotubes or vice versa.

6. PLOS authors have the option to publish the peer review history of their article (what does this mean?). If published, this will include your full peer review and any attached files.

Reviewer #1: No

Reviewer #2: No

---

## [Author Response · Author response to Decision Letter 0]

1 Jun 2024

＊＊＊＊＊＊＊＊＊＊＊＊＊＊＊

Point-to-point response to Reviewers’ comments:

Reviewer #1: A straightforward study examining the similarities between myotubes derived from myoblasts from inclusion body myositis patients and age matched controls. These samples are assessed by RNA-seq and compared to RNA-seq data from whole muscle biopsies. Overall, the study is interesting but several areas would strengthen the conclusions and allows for more complete interpretation of the data.

Methods: Central to this study is the isolation of myogenic precursor cells from human muscle biopsies. The protocol used (FACS sorting) relies heavily on negative sorts to isolate myoblasts (MuSCs?) and only CD56 is used as a positive sort. To increase confidence, the isolated cells should also be subjected to a positive sort, or in this case, since the RNA-seq analysis is complete, that validation of the purity of the cells used for differentiation be used with accepted markers for human MuScs (eg. Integrin alpha 7, m-cadherin, CD82).

Thank you for your valuable insights. As you mentioned, the myoblasts in this study were purified using FACS sorting, with negative sorting for CD45, CD31, and CD11b. We have included a figure depicting the gating strategy we previously reported as below (Koide et al.). In this strategy, we consistently create negative controls and set gating for CD56 positivity to accurately select myogenic precursors. Additionally, the high purity of differentiated myotubes is demonstrated in the revised Figure 1B-E as mentioned later.

Ref: Koide et al. Tohoku J Exp Med 2018

Furthermore, in response to reviewer comments, we analyzed the expression levels of muscle markers in myotubes using RNA-seq (S1 Fig). As suggested, the expression of human muscle stem cell markers such as Integrin alpha 7 (ITGA7), m-cadherin (CDH15), and CD82 was found to be high. Conversely, the expression of inflammatory, chemokine, and mesodermal markers such as Interleukin-1a (IL1A), C-C chemokine receptor type 2 (CCR2), and SRY-related HMG-box (SOX)10 was not detected.

These data further support the assertion that the cells isolated in this study are human myoblasts.

In the Results section, the following sentence has been added: “In both control and sIBM-derived myotubes, the expression of human muscle stem cell markers such as Integrin alpha 7 (ITGA7), m-cadherin (CDH15), and CD82 was observed to be high (S1 Fig).”

Figure 1B-E. Overlay of the DAPI and desmin stain is critical to appreciate the % of cells that have not differentiated. While this appears to be high for both conditions, a calculation of the differentiation index would also help appreciate the level of contamination from reserve cells and the expected variability from sample to sample and experiment to experiment. This has an impact on the downstream analysis and the interpretation of the findings.

For Figures 1B-E, it's notable that even instances with low fluorescence intensity are positive for Desmin. Following reviewer’s recommendation, including overlay images, we labeled them as Figure 1B-G. Quantification revealed that 95-98% of nuclei overlapped with Desmin (new Figure 1H). Both Control and sIBM samples exhibited similar positivity rates, indicating minimal impact on differentiation efficiency. This positivity rate aligns with previous reports (Oikawa et al.).

We added the sentences in Result section as below. “When quantified, it was determined that 95-98% of the nuclei exhibited colocalization with Desmin, as depicted in Figure 1H.”

Figure 2. For the RNA-seq analysis, it would be beneficial to include the FDR cut-offs used for significance in the text. Changes in the expression of factors associated with differentiation and myofiber maturity/contraction can be related to differences in the efficiency of differentiation between samples.

The False Discovery Rate (FDR) cutoff value was set at 0.05.　As we mentioned above, we analyzed the expression levels of muscle markers in myotubes using RNA-seq (new S1 Fig). As suggested, the expression of human muscle stem cell markers such as Integrin alpha 7 (ITGA7), m-cadherin (CDH15), and CD82 was found to be high. Conversely, the expression of inflammatory, chemokine, and mesodermal markers such as Interleukin-1a (IL1A), C-C chemokine receptor type 2 (CCR2), and SRY-related HMG-box (SOX)10 was not detected. These data further support the assertion that the cells isolated in this study are human myoblasts.

For the comparative analysis, it is not surprising that genes that are uniquely expressed in myofibers are also commonly regulated in whole muscle samples. More interesting however, is the massive number of genes oppositely regulated. No further analysis is provided here, but would be very useful in terms of understanding the origins. Was a similar analysis done for controls (or were the DEGs determined as relative to controls?). It would be interesting to see how divergent the control myotubes are from the healthy whole muscle, as a way to infer differences that the microenvironment contributes.

For the comparison between myotubes generated in culture and whole muscle biopsy samples, The gene name for Tmem8c is actually MYMK and this should be changed throughout.

Thank you for highlighting these important points. As you mentioned, the observation of genes expressed in myofibers but showing opposite expression patterns in patient muscle tissue suggests the involvement of different cell types. Given the pathology of sIBM involving inflammatory cell infiltration, fibrosis, and adipose tissues, potential confounding effects of non-muscle cells in expression data analysis are expected. We have addressed this point in the Discussion section.

While it is crucial to understand the extent of differences between Control Myotubes and healthy muscle, it was challenging to address this in the current study as Disease control Whole muscle samples were used. However, we acknowledge the importance of this aspect and consider it as a future direction, particularly through single-cell analysis to dissect the cellular resources of expression changes.

We added the description in the discussion session as follows: “This change is not observed in myotubes. This activation might represent a non-cell-autonomous mechanism contributing to muscle degeneration in sIBM. It may simply be expressed at low levels in myotubes but not reaching the threshold for detection, which is a limitation of our analysis in this study. Discriminating between cell types, such as through single-cell RNA-seq analysis, is a task for future research.”

Furthermore, TMEM8C has been renamed as MYMK, as indicated in the revised figure and text.

Figure 3. The text says that the levels were verified at the protein level, however FPKM are shown and no quantification over the patient samples are provided for the immunostained images. Quantifications are essential to validate the conclusions.

The reviewer’s comments are absolutely valid. We have re-quantified the immunostaining and created a graph. A statistically significant difference was found between the control and sIBM (p=0.0358). The number of muscle fibers expressing TMEM8C/MYMK is shown as a percentage. We have also added the quantitative data to Figure 3D.

Figure 4. Similar comments to Figure 3.

Thank you for pointing this out. We have also quantified the ApoE and amyloid oligomer, and the results have been added to Figure 4. Significant differences were found in both cases.

Discussion:

Line 323. I do not think the word pathogenesis applied to the myotubes in culture. I appreciate that you mean a myofiber-intrinsic contribution to the pathogenesis of the disease, but this is not clear as written.

Thank you for pointing out that the term 'pathogenesis' may not be appropriate in the context of myotubes in culture. Following your suggestion, I have revised the phrase to 'myofiber-intrinsic contribution to the pathogenesis of the disease' for clarity.

This statement (lines 332-336) “The examination of muscle biopsies suggested altered gene expression in lymphocytes and other immunopathological conditions, phenomena not mirrored in myotube samples.” Is not supported by the data shown.

The statement in lines 337-340 is also unclear as to where the findings come from (as related to SASP). Please clarify.

The description regarding STING did not extend beyond speculation. As it exceeded the scope of this paper, it has been removed. The points raised by the reviewer have been addressed as limitations, and the paragraph has been revised as follows: 

“Significantly, members of the interferon-2 pathway, such as PSMB8, GBP1, and GBP2, were elevated in biopsy samples, aligning with prior analyses of biopsied muscle31. This change is not observed in myotubes. This activation might represent a non-cell-autonomous mechanism contributing to muscle degeneration in sIBM. It may simply be expressed at low levels in myotubes but not reaching the threshold for detection, which is a limitation of our analysis in this study. Discriminating between cell types, such as through single-cell RNA-seq analysis, is a task for future research.”

Minor comments: Care should be taken to use the appropriate gene nomenclature for humans throughout to avoid confusion with mouse genes or protein names.

I capitalized the names of human genes.

Reviewer #2: Suzuki et al. present an interesting finding regarding the pathological mechanism(s) involved in sIBM. Previous findings have failed to define the pathology as autoimmune or degenerative, not having determined whether the pathology observed is primarily due to intrinsic or extrinsic forces. In their study the authors confront the gene expression profile using RNA-seq between sIBM biopsy samples and controls as well as with myotubes derived from myoblasts isolated from control and sIBM biopsies. In such a manner, the study is able to differentiate between gene expression profiles of sIBM biopsies (due both to intrinsic and extrinsic forces), and gene expression profiles of in vitro differentiated sIBM-derived myotubes (due to intrinsic forces). The data are interesting and find similar results to previous reports. What is missing is a full consideration of the findings.

Major comments:

1. Figure 1, please show the merged image of DAPI and desmin.

For Figures 1B-E, it's notable that even instances with low fluorescence intensity are positive for Desmin. Following reviewer’s recommendation, including overlay images, we labeled them as Figure 1B-G. Quantification revealed that 95-98% of nuclei overlapped with Desmin (new Figure 1H). Both Control and sIBM samples exhibited similar positivity rates, indicating minimal impact on differentiation efficiency. This positivity rate aligns with previous reports (Oikawa et al.).

We added the sentences in Result section as below. “When quantified, it was determined that 95-98% of the nuclei exhibited colocalization with Desmin, as depicted in Figure 1H.”

2. As sIBM leads to muscle death and atrophy, it is important to known if any difference was observed in the differentiation potential/rate and viability between controls and sIBM biopsy-derived myoblasts.

Thank you for your valuable insights. The high purity of differentiated myotubes is demonstrated in the revised Figure 1B-E. Furthermore, in response to reviewer1’s comments, we analyzed the expression levels of muscle markers in myotubes using RNA-seq (Supplemental Fig). As suggested, the expression of human muscle stem cell markers such as Integrin alpha 7 (ITGA7), m-cadherin (CDH15), and CD82 was found to be high. Conversely, the expression of inflammatory, chemokine, and mesodermal markers such as Interleukin-1a (IL1A), C-C chemokine receptor type 2 (CCR2), and SRY-related HMG-box (SOX)10 was not detected.

These data further support the assertion that the cells isolated in this study are human myoblasts.

In the Results section, the following sentence has been added: In both control and sIBM-derived myotubes, the expression of human muscle stem cell markers such as Integrin alpha 7 (ITGA7), m-cadherin (CDH15), and CD82 was observed to be high (S1 Fig).

If stressors such as BSO are not applied, the survival rate of sIBM myoblasts does not differ from that of the control, as reported by Oikawa et al.

Ref: Oikawa Y, et al. PLoS One. 2020;15:e0231064.

3. Data from myotubes suggests a potential alteration of the insulin growth factor response and cellular adhesion in sIBM. How does this fit in with previous findings in the field?

Thank you for highlighting this important point. Regarding insulin-like growth factor (IGF), Broccolini et al.'s pathological work indicates increased expression in sIBM. They also conducted experiments where normal primary muscle cultures were stimulated for 24 hours with the Amyloid beta peptide, which corresponds to the biologically active domain of Amyloid beta. In their discussion, they propose that IGF-I overexpression may represent a reactive response to Amyloid beta toxicity. This is intriguing in the context of our observed increase in A beta expression.

Added to the Discussion section:

“Additionally, it is possible that Amyloid beta overexpression might influence the characteristic KLRG1-T cells of inclusion body myositis via the insulin-like growth factor signaling pathway or immune modulation mechanisms.”

Ref: Broccolini A, et al. J Neuropathol Exp Neurol. 2004;63:650-9.

Ref: Laskowitz DT, et al. J Lipid Res. 2000;41:613-20.

Ref: Goyal NA, et al. Neurology. 2022;98:e1374-e1383.

4. In discussing differences between data in Supplementary Table 2 and Supplementary Table 3, the authors observe the absence of immunopathological gene expression (lines 333-341) in myotubes and suggest that some phenotypic findings may be linked to STING and non-cell-autonomous mechanisms. The authors should be careful here with this interpretation. First, there is more than STING to consider; second, data in Supplemental Table 2 could suggest a defective type I interferon response; and third, the authors used a cut-off for choosing which genes to include in their analysis, but it should be noted that in almost all cases, the ratio of gene expression of common genes between sIBM-derived myotubes/control versus sIGM biopsy/control is always decreased. therefore some changes seen in the biopsy samples may have been below the threshold for consideration in the myotubes, but none-the-less still present.

The description regarding STING did not extend beyond speculation. As it exceeded the scope of this paper, it has been removed. The points raised by the reviewer have been addressed as limitations, and the paragraph has been revised as follows: 

“Significantly, members of the interferon-2 pathway, such as PSMB8, GBP1, and GBP2, were elevated in biopsy samples, aligning with prior analyses of biopsied muscle31. This change is not observed in myotubes. This activation might represent a non-cell-autonomous mechanism contributing to muscle degeneration in sIBM. It may simply be expressed at low levels in myotubes but not reaching the threshold for detection, which is a limitation of our analysis in this study. Discriminating between cell types, such as through single-cell RNA-seq analysis, is a task for future research.”

5. The authors focus then on ApoE, TMEM8C and myogenin. The pathophysiological role that upregulation of these proteins has in sIBM is difficult to determine. TMEM8C and myogenin should, at face value, favor muscle development. Likewise, the role of ApoE is difficult to determine due to its varying immunomodulatory roles, including T cell suppression.

The observed changes in TMEM8C and myogenin indicate muscle development and likely reflect the muscle damage and repair processes in sIBM. It is hypothesized that there may be mechanisms disrupting the remodeling process, resulting in impaired muscle regeneration. Moreover, ApoE plays immunomodulatory roles, including T cell suppression, and its abnormalities could potentially affect KLRG1-positive T cells. These points remain speculative, but we have mentioned their relationship with Aβ briefly in the text, considering them important topics for future analysis.

Added to

---

## [Decision Letter · Decision Letter 1]

11 Jun 2024

Sporadic inclusion body myositis-derived myotube culture revealed muscle cell-autonomous expression profiles

PONE-D-24-02730R1

Dear Dr. Suzuki,

We’re pleased to inform you that your manuscript has been judged scientifically suitable for publication and will be formally accepted for publication once it meets all outstanding technical requirements.

Kind regards,

Keisuke Hitachi

Academic Editor

PLOS ONE

Additional Editor Comments (optional):

Reviewers' comments:

Reviewer's Responses to Questions

**Comments to the Author**

1. If the authors have adequately addressed your comments raised in a previous round of review and you feel that this manuscript is now acceptable for publication, you may indicate that here to bypass the “Comments to the Author” section, enter your conflict of interest statement in the “Confidential to Editor” section, and submit your "Accept" recommendation.

Reviewer #1: All comments have been addressed

Reviewer #2: All comments have been addressed

2. Is the manuscript technically sound, and do the data support the conclusions?

Reviewer #1: Yes

Reviewer #2: Yes

3. Has the statistical analysis been performed appropriately and rigorously? 

Reviewer #1: Yes

Reviewer #2: Yes

4. Have the authors made all data underlying the findings in their manuscript fully available?

Reviewer #1: Yes

Reviewer #2: Yes

5. Is the manuscript presented in an intelligible fashion and written in standard English?

Reviewer #1: Yes

Reviewer #2: Yes

6. Review Comments to the Author

Reviewer #1: The authors have carefully and completely addressed my comments from the first round of review. The revised manuscript better discusses limitations of the experimental approach and also provides additional information to the reader which helps interpretation of the data sets.

Reviewer #2: (No Response)

7. PLOS authors have the option to publish the peer review history of their article (what does this mean?). If published, this will include your full peer review and any attached files.

Reviewer #1: No

Reviewer #2: No

---

## [Editor Report · Acceptance letter]

27 Jun 2024

PONE-D-24-02730R1 

PLOS ONE

Dear Dr. Suzuki, 

I'm pleased to inform you that your manuscript has been deemed suitable for publication in PLOS ONE. Congratulations! Your manuscript is now being handed over to our production team.

Kind regards, 

on behalf of

Dr. Keisuke Hitachi 

Academic Editor

PLOS ONE